# Role of FGF15 in Hepatic Surgery in the Presence of Tumorigenesis: Dr. Jekyll or Mr. Hyde?

**DOI:** 10.3390/cells10061421

**Published:** 2021-06-07

**Authors:** Albert Caballeria-Casals, Marc Micó-Carnero, Carlos Rojano-Alfonso, Cristina Maroto-Serrat, Araní Casillas-Ramírez, Ana I. Álvarez-Mercado, Jordi Gracia-Sancho, Carmen Peralta

**Affiliations:** 1Institut d’Investigacions Biomèdiques August Pi i Sunyer (IDIBAPS), 08036 Barcelona, Spain; acabalca31@alumnes.ub.edu (A.C.-C.); mico@clinic.cat (M.M.-C.); rojano@clinic.cat (C.R.-A.); 2Aplicacions i Muntatge Torelló SL (AMT), 08570 Torelló, Spain; cristina.maroto@uvic.cat; 3Hospital Regional de Alta Especialidad de Ciudad Victoria “Bicentenario 2010”, Ciudad Victoria 87087, Mexico; acasillas@docentes.uat.edu.mx; 4Facultad de Medicina e Ingeniería en Sistemas Computacionales de Matamoros, Universidad Autónoma de Tamaulipas, Matamoros 87300, Mexico; 5Departamento de Bioquímica y Biología Molecular II, Escuela de Farmacia, Universidad de Granada, 18071 Granada, Spain; alvarezmercado@ugr.es; 6Institute of Nutrition and Food Technology “José Mataix”, Center of Biomedical Research, University of Granada, 18016 Armilla, Spain; 7Instituto de Investigación Biosanitaria ibs.GRANADA, Complejo Hospitalario Universitario de Granada, 18014 Granada, Spain; 8Liver Vascular Biology Research Group, Barcelona Hepatic Hemodynamic Laboratory IDIBAPS, 03036 Barcelona, Spain; 9Centro de Investigación Biomédica en Red de Enfermedades Hepáticas y Digestivas (CIBERehd), 08036 Barcelona, Spain

**Keywords:** hepatocellular carcinoma, liver transplantation, ischemia-reperfusion injury, liver surgery, fibroblast growth factor

## Abstract

The pro-tumorigenic activity of fibroblast growth factor (FGF) 19 (FGF15 in its rodent orthologue) in hepatocellular carcinoma (HCC), as well as the unsolved problem that ischemia-reperfusion (IR) injury supposes in liver surgeries, are well known. However, it has been shown that FGF15 administration protects against liver damage and regenerative failure in liver transplantation (LT) from brain-dead donors without tumor signals, providing a benefit in avoiding IR injury. The protection provided by FGF15/19 is due to its anti-apoptotic and pro-regenerative properties, which make this molecule a potentially beneficial or harmful factor, depending on the disease. In the present review, we describe the preclinical models currently available to understand the signaling pathways responsible for the apparent controversial effects of FGF15/19 in the liver (to repair a damaged liver or to promote tumorigenesis). As well, we study the potential pharmacological use that has the activation or inhibition of FGF15/19 pathways depending on the disease to be treated. We also discuss whether FGF15/19 non-pro-tumorigenic variants, which have been developed for the treatment of liver diseases, might be promising approaches in the surgery of hepatic resections and LT using healthy livers and livers from extended-criteria donors.

## 1. Introduction

Currently, liver transplantation (LT) remains an unsolved problem in clinical practice, not only due to the lack of donor grafts but also because of the risk factors of liver dysfunction or failure that show steatotic livers (present in 30% of total liver grafts) [1] or livers from a brain-dead (BD) donor (the 80% of deceased donors) [2]. BD negatively affects the hepatic function following transplantation [3,4], and livers with steatosis are more susceptible to ischemia-reperfusion (IR) injury, thus negatively affecting liver function and graft quality [5]. Therefore, many liver grafts are discarded, exacerbating the shortage of grafts [2,6]. In the surgery of hepatic resections, IR injury is currently performed to avoid excessive bleeding [7]. On the other hand, IR injury negatively affects regenerative capacity after partial hepatectomy (PH) [6], particularly in steatotic livers, resulting in the worst postoperative outcomes [6,8].

The role of FGF15/19 in metabolism is well known. After feeding, gallbladder accumulated bile acids (BA) are released into the intestine due to gallbladder contraction [9]. In response to the activation of pathways stimulated by the binding of BA to its receptor farnesoid X receptor (FXR), ileal enterocytes secrete fibroblast growth factor (FGF) 19 (FGF15 in its rodent orthologue) into the portal system [10], by which it reaches the liver. Once in the liver, FGF15/19 negatively regulates BA synthesis [11] and also stimulates gallbladder refilling through a process that ends with gallbladder smooth muscle relaxation [9]. Although considered the main one, this is not the only regulatory function of FGF15/19. Several studies have pointed out the role of FGF15/19 as a postprandial regulator of glucose and protein homeostasis [12,13]. Moreover, it has been reported that FGF15/19 also exerts biological effects on the adipose tissue and brain [14,15]. The important metabolic role of FGF15/19 is reflected by the following: the dysregulation of FGF19 has been associated with inflammatory bowel disease [16], type II diabetes, obesity [17], non-alcoholic fatty liver disease (NAFLD) [18], and in a wide range of cholestasis spectrum [19]. Thus, FGF15/19 might be considered as a therapeutic target for different metabolic diseases.

Recent data indicate that BD induces gut inflammation, thus affecting the FGF15/19 secretion [20]. Through its hepatic receptor fibroblast growth factor receptor 4 (FGFR4), the administration of FGF15 in rat steatotic and non-steatotic LT from BD donors controls BAs secretion by regulating the cytochrome P457A1 (CYP7A1) [20]. Moreover, exogenous FGF15 improves liver regeneration by activating Hippo/Yes-associated protein (YAP) pathway [20,21]. This downstream signaling pathway of FGF15 (namely Hippo-YAP) seems to be highly beneficial in surgeries that require liver regeneration as LT or PH [20,21,22,23]. However, of scientific and clinical interest, many studies have reported the negative effects of the YAP pathway (activated by FGF15/19) in hepatocellular carcinoma (HCC) and other cancers [24,25]. Thus, FGF15/19 and its downstream signaling pathways have two faces dependently on the pathological conditions, for instance, the presence or absence of tumorigenesis. In this manuscript, the dual effect of FGF15/19 will be reviewed. We will discuss how non-pro-tumorigenic variants of FGF15/19 would be a promising approach to modulate IR injury and regenerative failure in PH and LT surgeries. Such drugs consist of engineered FGF15/19-like molecules that mimic FGF15/19 function. In addition, we will describe the current preclinical models available showing the different pathways potentially responsible for the pro/anti-tumorigenic activities of FGF15/19.

## 2. Role of FGF15/19 in Liver Tumorigenesis

It is widely known that some liver aggressions such as hepatitis B and C virus infections, sustained alcohol ingestion, exposure to environmental toxins, and NAFLD cause sustained inflammation, which, in turn, can evolve into fibrosis, cirrhosis, and, finally, a background of chronic liver injury, which easily ends up in some type of liver cancer, especially HCC and cholangiocarcinoma [26,27,28,29]. Different results have been reported on the role of FGF15/19 and their interaction with FGFR4, its receptor, in liver tumorigenesis [30]. Indeed, FGF15/19 can act by promoting or avoiding HCC, results that are completely opposite [31,32,33,34,35]. So, because these inconsistencies exist, in our view, before analyzing whether FGF19 can be a potential target to protect against damage and regenerative failure associated with LT and hepatic resections, the reasons that explain why FGF19 might act as a protective or tumorigenic agent are required. This is what we have done in the current review.

The involvement in FGF15/19 and/or FGFR4 in cancer has been extensively reported, but the pathways and mechanisms by which FGF15/19 and/or FGFR4 affect tumorigenesis remains elusive. Such aspects will be re-organized in detail in the following paragraphs in order to have a more global view of this whole process. As we will develop below, however, it is possible to point out that the contribution of FGF15/19 in cancer (either to potentiate or reduce it) may be related to the inflammation process, control of cell adhesion molecules, proliferation and apoptosis, regulation of BA toxicity, its relationship with other growth factors (such as vascular endothelial growth factor receptor (VEGFR), with effects on angiogenesis), and with different genetic and epigenetic alterations.

### 2.1. Involvement of BA in the Effects of FGF15/19 on Tumorigenesis

Like other growth factors, FGF19 has potent effects on cell proliferation, survival, and motility, enhancing or reducing these processes in response to different stimuli [36]. Due to the physiological FGF19 function as an enterohepatic hormone regulating BA synthesis in intermediary metabolism [37], and the high expression of FGFR4 and β-klotho in hepatocytes [26], it is clear that deficiencies in FGF19 synthesis or down-regulation in FGFR4 and β-klotho will have a significant impact on the liver. Given the relationship between FGF15/19 and BA, it is important to analyze the role of BAs in the development of tumorigenesis. BAs can cause or alter some processes that can potentially cause cancer, such as DNA damage and genomic instability associated with oxidative stress [38], apoptosis [39], epigenetics [40], or modification of gut microbiota [41]. Moreover, studies in rats have found that BAs can act as mitogenic agents in colon epithelial cells [42], reduce apoptosis [43], and promote colorectal cancer formation and progression [44]. This link between BAs (specifically with deoxycholic acid (DCA)) and colorectal cancer has been known since 1940 [45], but the intestine is not the only organ where the presence of BAs has been linked to cancer. In fact, hydrophobic BAs, among which is DCA, are highly cytotoxic and can induce hepatocyte damage [46], specifically mitochondrial damage, cell membrane disruption, increasing levels of reactive oxygen species, and sustained inflammation, which leads to the development of HCC if BA levels are not drastically reduced [47,48]. This negative impact of BA in hepatocytes can be mitigated by the Hippo-YAP pathway. Indeed, BA accumulation in hepatocytes induces the overexpression of IQ motif containing GTPase activating protein 1 (IQGAP1), which decreases cell-cell adhesion dissociating α-catenin from E-cadherin-β-catenin complex, that cause the translocation of YAP into the nucleus where activates protective mechanisms [49].

After the liver injury produced by a loss of liver mass (such as in a PH), BAs levels increase drastically within the liver, provoking the activation of the mechanisms explained above, increasing liver regeneration and decreasing apoptosis, thus provoking liver tissue repair [20]. By contrast, the dysregulation in the BA-Hippo-YAP pathway could result in the initiation of a tumorigenic process [24,25,50,51]. With the aim of avoiding it, the high levels of hepatic BAs might result in high levels of intestinal BAs that will be bind to FXR of the enterocytes inducing the production of FGF15, which will be released into the portal circulation. The binding of FGF15 with FGFR4 in the gut triggers two facts. On the one hand, the expression of CYP7A1 is blocked, and, consequently, the production of new BAs stops. On the other hand, when FGFR4 is active because of binding to FGF15, it phosphorylates and activates neurofibromin 2 (NF2), which in turn activates mammalian STE20-like protein kinase (Mst1/2) and the consequent Hippo pathway that provokes YAP phosphorylation. Its phosphorylation causes its sequestration in the cytoplasm, finally provoking its degradation, thus inhibiting pro-proliferative and anti-apoptotic signals avoiding the risk of tumorigenesis [52]. Giving all this data in mind, the authors of that study consider that NF2 is acting as a switch in the regeneration process, allowing it or not depending on the levels of BAs and the presence of FGF15. This system would allow the liver to begin the process of regeneration and stop it when the size of the remaining liver following PH has recovered. However, more preclinical and clinical studies are required in order to clearly elucidate the exact mechanism by which full recovery of liver mass is detected. In our view, for the appropriate reparation of the damaged liver (liver regeneration) following PH or LT, it is required a regulation between BA-FGF15/FGFR4-NF2-Hippo-YAP. What is clear is that any disruption of this pathway (for instance, that affecting NF2, as the authors themselves already indicate in their study) is a potential risk factor for HCC.

### 2.2. Involvement of FGF15/19 and Inflammation in the Hepatic Microenvironment

As mentioned before, the liver injury caused by diverse aggressions alters the normal hepatic microenvironment and generates inflammation, necrosis, and regeneration, which change to favor the transformation of selected hepatocyte populations into dysplastic nodules and easily evolve to liver cancer [28]. This is consistent with several works pointing out that FGFRs signaling abnormalities observed in liver cancer can contribute to the mechanisms involved in liver tumorigenesis mediated by cirrhosis [53]. In fact, as previously reported by Feng et al., inflammation plays an important role in the hepatic microenvironment alterations (such as fibrosis and cirrhosis) [54], and other reports indicate that some cytokines such as tumor necrosis factor (TNF), interleukin 1 (IL-1), interleukin 6 (IL-6), interleukin 10 (IL-10), chemokine (C-C motif) ligand 2 (CCL2) and other inflammatory factors such as macrophage colony-stimulating factor (M-CSF) or VEGFR can regulate the tumor microenvironment. Under these conditions, FGFR4 can potentiate cancer progression by modulating nuclear factor kappa B (NF-κB) and increasing the inflammatory microenvironment, thus entering in a loop that promotes tumor development. Moreover, it is known that under oxidative stress associated with inflammatory processes, FGFR4 can regulate cancer cell survival via mitogen-activated protein kinase (MAPK) and phosphoinositide 3-kinase/protein kinase B (PI3K/AKT) signaling pathways (Figure 1) [36,55].

Cytokines, kinases, and transcription factors contribute to tumor progression in different experimental models of cancer [56]. It should be considered that they also play a crucial role in hepatic IR injury associated with PH as well as in LT by activating survival pathways to reduce damage and resolve inflammation [57]. Such observations are derived from experimental models of IR in the absence of cancer and liver regeneration [58,59]. In addition, the role of cytokines and growth factors such as TNF, IL-1, IL-6, VEGFR, and the nuclear factor NF-κB have been considered crucial in the regenerative and cell repairing process in experimental models of PH and LT in the absence of tumorigenesis [8,60]. Nevertheless, the exact role of such mediators when liver surgeries are performed and in the presence of tumorigenesis needs to be clarified. It should be considered that IR is commonly performed in the PH of tumors to avoid bleeding, but it promotes inflammation and damage and negatively affects the regenerative process [61]. In addition, when LT is performed due to the presence of HCC, such mediators mentioned above (cytokines, kinases, and transcription factors) might be released to the circulation of the recipient contributing to tumor progression. Then, when “healthy” liver grafts are implanted in the recipient, they might be uptaken by the liver from the circulation, affecting the postoperative outcomes following LT. In addition, during PH, it should be considered not only the uptake of such circulating mediators by the remaining liver, but these might remain present after tumor resection in the remaining liver due to undetected and unresected micrometastasis [62]. Thus, these inflammatory mediators might favor tumorigenesis but might also exert beneficial effects on regeneration and hepatic damage. If this occurs in “healthy” liver grafts, such effects might be also more evidenced in extended-criteria donor grafts, such as steatotic livers, because this type of liver shows more vulnerability to IR damage and regenerative failure when compared with the results on non-steatotic ones [21]. Consequently, under such conditions, it is difficult to determine postoperative outcomes.

As the interaction of FGFR4 with its ligand, FGF19, affects the downstream signaling pathway, it is obvious that high expression of FGF19 and/or FGFR4 is correlated with a poor prognosis in HCC cancer [63]. There is evidence in human colon cancer cell lines, colon tumor xenografts, and FGF19 transgenic mice that blocking selectively the interaction of FGF19 and FGFR4 using an anti-FGF19 monoclonal antibody has a potential pharmacological anti-tumorigenic effect [64]. Another study in human colon cancer cell lines indicates that when FGF19 interacts with its receptor FGFR4, there is an increase in phosphorylated glycogen synthase kinase 3β (GSK3β), an intermediary of Wnt pathway, which actives β-catenin, that, in turn, causes loss of β-catenin-E-cadherin binding [65]. This loss is associated with the disruption of the epithelial barrier and leads to a wide variety of human malignancies (Figure 1) [66]. Moreover, data obtained from a model of co-culture of the colorectal cancer cell line with tumor-associated fibroblasts (TAF) indicate that FGF19-FGFR4-mediated β-catenin-phosphorylation is associated with the metastatic process. In that study, it was observed that when the FGF19- FGFR4 interaction occurs in TAF, FGFR4 can phosphorylate β-catenin, which, in turn, can be translocated into the nucleus where can mediate the tumor-stroma interactions, facilitating the metastatic process [67]. However, this effect was abrogated when the FGF19-FGFR4 interaction was disrupted using an antibody against anti-FGF19 [65]. In addition, studies based on mouse models indicate that individual overexpression of FGF19 and/or FGFR4 are not the only risk factors for HCC and other cancer types since FGFR4 hyperactivation mediated by abnormal FGF19 expression and, consequently, signaling is just an important issue [68]. Moreover, Cheng et al., based on a previous study in models of transgenic mice [69], suggest that FGF-FGFR signaling contributes to endothelial cell differentiation signaling, a process that is essential in normal angiogenesis and blood vessel formation, whereas alterations in such axis are important for tumor-associated neoangiogenesis [53]. In the same line, upregulation of FGFR4 expression has also been shown to promote resistance to chemotherapy [70].

### 2.3. FGF15/19: Prognostic Factor of HCC?

At present, there are no appropriate biochemical markers to determine the degree of fatty infiltration in the liver to estimate the prognostic following PH or LT. Several reports have demonstrated that liver steatosis is associated with the elevation of certain clinical and biochemical markers [71,72,73] with low predictive potential. Indeed, transaminases and specifically alanine aminotransferase (ALT), commonly used as liver damage markers, are not 100% reliable because some patients suffering from NAFLD do not present high levels of ALT [74]. Recent studies evaluating other biomarkers, including circulating miRNA, reported promising results [75], although larger studies are still necessary to validate them in the future. According to existing data, it is difficult to elucidate whether the levels of FGF19 and/or FGFR4 are indicative of HCC progression and/or the progression of the remaining liver following PH of tumors as well as in livers grafts implanted in the recipient after removing liver with HCC. It is even possible that they are not indicative of none of both aspects since FGF19 and/or FGFR4 levels are affected by both processes (by HCC and by the surgery itself).

### 2.4. Mechanisms of Action of FGF15/19-FGFR4

FGFRs usually act through two main pathways: modulating hepatocyte proliferation and controlling the apoptotic processes. In the specific case of FGFR4, it needs to form a heterodimer with another FGFR4 and with β-klotho to be activated [31]. In these conditions, when FGF19 binds to FGFR4 generates a conformational change in both FGFR4 receptors, and then FGFR4 is phosphorylated and activated. In turn, FGFR4 phosphorylates FGF receptor substrate 2 (FRS2) and recruits the adaptor molecule growth factor receptor-bound protein 2 (GRB2). This complex activates two major signaling pathways: the Ras-Raf-ERK1/2-MAPK, which leads to cell proliferation, and the PI3K-AKT, which has anti-apoptotic effects (Figure 1) [34]. These mechanisms of action explain the regenerative effects of the FGF19-FGFR4 axis. The crucial role of FGFR4 downstream signaling pathways in reducing IR injury and promoting cell damage repair has been previously demonstrated in steatotic and non-steatotic livers undergoing either PH or LT [21]. However, it has to be considered that activation of such signaling pathways (Ras-Raf-ERK1/2-MAPK and PI3K-AKT) in tumor cells may have detrimental consequences since it would result in tumorigenesis exacerbation. So, it is expected that overexpression of FGF19, β-klotho, and/or FGFR4 in tumor cells is related to the development and progression of HCC [76,77]. This opens new questions on the role of such signaling pathways in the remaining liver following PH of tumors as well as in livers grafts implanted in the recipient after removing liver with HCC. Under these conditions, the activation of such signaling pathways would protect against IR damage and regenerative failure but would also promote the prospect of the tumorigenic process. On the other hand, other results indicate that the absence of FGFR4 accelerates the progression of chemical-induced HCC, and hence FGFR4 has an important anti-proliferative effect [31,34], and the down-regulation of FGFR4 is related to a higher risk of hepatotumorigenesis [32]. Altogether, these data indicate that FGFR4 activation is related to tumor progression suppression and with the protection of steatotic and non-steatotic livers against damage and regenerative failure [21]. Thus, FGFR4 activation would be beneficial for all processes against IR damage, regenerative failure, and hepatic tumor progression.

We believe that the controversial data on the pro or anti-tumorigenic actions of the FGF19-FGFR4 axis can be explained, at least partially, by the following. The differences in the experimental models, the different ways of cancer induction, and the types of pharmacological or transgenic modulations used to modulate the FGF19-FGFR4 actions are relevant and must be considered. The studies that confirmed the beneficial effects of FGFR4 on tumorigenesis have mainly been based on the use of murine tumorigenesis models induced by chemicals such as dimethylnitrosamine (DEN) or carbon tetrachloride (CCl_4_). However, the studies that indicated the pro-tumorigenic actions of FGF19-FGFR4 are mostly based on in vitro models, xenograft, or transgenic animals. Regarding in vitro studies, it should be considered that in vitro results cannot always be extrapolated to in vivo models. In addition, cancer cells from human colon cancer have been used in the mentioned studies, but such cell types might present differences with respect to human HCC-derived cells. In relation to the transgenic mouse models, authors used genetically modified mice constitutively expressing the human *fgf19* gene [78]. However, in such conditions, the actions of FGF19 in mice might be different from those observed in humans. This may be the case of xenograft models, where both mouse and human elements coexist. The in vivo chemically-induced HCC models (DEN and CCl_4_) are currently selected to eliminate or downregulate the FGFR4 receptor before proceeding to cancer induction. Depending on the dose and/or the pre-treatment time of these chemical compounds (DEN and CCl_4_), they might induce fibrosis, cirrhosis, or hepatocarcinoma [79,80]. The interpretation of the results obtained from such experimental models might be difficult. Indeed, the exacerbated tumorigenic effects observed in such animals (exposed to FGFR4 down-regulation and chemical compounds) compared with those only treated with chemical compounds might not be exclusively explained by the beneficial effect of FGFR4 on tumorigenesis. The hepatic damage induced by FGR4 down-regulation should also be considered: it is well known that one of the main physiological functions of FGFR4 is to control liver BA levels to avoid its injurious effects on liver functionality. Thus, liver damage might be present in mice with FGFR4 down-regulation induced genetically, even before the hepatic damage induced by the chemical exposition. In such conditions, the hepatic damage will be exacerbated since the induced cancer would be much more severe or aggressive. Consequently, future studies with much better models of cancer induction and cell types to simulate as much as possible the clinical conditions are required.

Another possible explanation for that change to the behavior of FGF19/FGFR4 is provided by Heinzle et al., who suggested that the tumor-suppressive function probably depends on the lineage-specific expression of β-klotho, which belongs to a protein family known for its role in anticancer processes [76]. They cite a work [32] where DEN-mediated HCC induction in FGFR4 knock-out mice promoted tumor growth while spontaneous tumor formation remained unaltered. The authors hypothesized that the tumor-suppressor function of FGFR4 could be the result of its co-expression with β-klotho. Accordingly, in an experimental model of FGFR4 or β-klotho deletion in mice, it has been described that the presence of β-klotho could attenuate the carcinogenic potential of the FGF19-FGFR4 interaction, so the anticancer character of β-klotho prevails over the FGFR4 one [31]. Another study on the effect of FGF21 (FGFR4 ligand) in chemically-induced hepatocarcinogenesis [81] can support such a hypothesis. Here, the authors observed that overexpression of FGF21 delays DEN-induced tumor formation in mice, and they attributed it to the activation of hepatocyte’s FGFR4 in an initial stage. Thus, FGFR4 would protect against cancer due to the interaction of FGFR4 with the β-klotho. Despite this, several other studies reported observations that are in clear discrepancy with this statement. Some of them, mainly reported in models of cancer cell lines or transgenic mice [63,82,83], indicate that the blockage of FGFR4 dimerization with antibodies or the use of FGFR4 knock-out mice completely prevented hepatocarcinogenesis [63]. Consequently, on the basis of the different results reported in the literature to date on the role of FGFR4 in tumorigenesis as well as their downstream signaling pathways, it is difficult to discern whether we should aim to inhibit or activate the FGF19/FGFR4 signaling pathway to protect against tumorigenesis and to reduce IR damage and regenerative failure in PH and LT. Nevertheless, in our view, further investigations focused on β-klotho might result in protective treatments for liver cancer progression and pathological problems associated with PH and LT (namely IR injury and regenerative failure).

Given the physiological functions of the FGF19-FGFR4 pathway in BA synthesis [84], the blockade of FGF19, FGFR4, or β-klotho, as a potential strategy to reduce the risk of cancer progression, would result in hepatic deregulation of BA synthesis. This is relevant because bile salts can elicit a hepatocellular proliferative response mediated by ileum-derived FGF15 according to preclinical and clinical data reported by Uriarte et al. in the mouse model [85]. Bile salts circulation deregulation can affect the tumorigenesis process. In fact, the pharmacological induction of RXR (the bile salts receptor) can derive hepatomegaly with the subsequent risk of cancer [35]. These results on deregulation of BA synthesis have also been reported in such surgical conditions and induce severe damage. In the remaining liver following PH of tumors as well as in livers grafts implanted in the recipient after removing liver with HCC, the deregulation of BA synthesis and the subsequent liver damage would be exacerbated.

### 2.5. Involvement of Genetic and Epigenetic FGF15/19 and FGFR4 Alterations in Tumorigenesis

In addition to all of the FGFR4 alterations due to the different kinds of interactions mentioned above, this receptor may suffer genetic alterations. Indeed, in many cancers, mutations, translocations and truncations of growth factors, cell cycle checkpoints, and other oncogenes are common genetic alterations with a deep impact in angiogenesis, invasion, metastasis, or response to therapy, among other effects [86]. The FGF19-FGFR4 axis is not an exception. There are some reports in human cancer cell lines about one specific *fgfr4* mutation with a relevant impact on tumorigenesis [86,87,88]. This is based on a single nucleotide polymorphism that implies the substitution of glycine by an arginine at the position 388 (G388R), which has been associated with reduced patient survival for head and neck carcinoma more aggressive variants of colorectal cancer, soft tissues carcinomas, prostate cancer, breast cancer, lung cancer, rhabdomyosarcoma, and HCC, caused mainly by an increment of local tumor growth and enhanced metastasis [36,83]. This substitution of one glycine by one arginine could be related to an increased FGFR4 mRNA expression in relation to normal tissue [77]. Moreover, data obtained in murine models indicate that FGFR4 overexpression caused by deregulation in gene transcription results in a ligand-independent transcription and/or activation of FGFR4 signaling [89]. Presta et al. described different FGFR4 genetic alterations in human cancer, including gene amplification, intragenic translocations, and chromosomal translocations that can result in receptor overexpression [84]. One of the most typical genetic alterations is the fusion of the tyrosine kinase motif with a transcription factor domain that leads to a permanent FGFR4 activation acting as an oncogene because signal transduction is constantly being performed without de receptor having any ligand to indicate so [90]. It should also be considered that the *histone deacetylase 2 (hdac2)* gene, a histone deacetylase, is overexpressed in tumor tissue of the worst prognostic subgroup of HCC patients, indicating that epigenetic alterations are just as important as the genetic ones [28]. The epigenetic regulation of FGF19 and FGFR4 is poorly investigated. In our view, intensive investigations on the possible relationship between HDAC2 and FGFR4 could be of scientific and clinical interest in hepatic tumorigenesis.

### 2.6. Role of FGFR4 on Metastasis

It is well known that in a situation of sustained accumulation of genetic and epigenetic alterations, the tumor microenvironment may undergo significant changes that lead to a higher risk of tumorigenesis or aggressiveness [28]. This is because, in the interface between tumor mass and healthy tissue, there are coexisting cells with different degrees of alteration that communicate with each other. In this reciprocal interaction between neoplastic and tumor-activated stroma cells, FGFs and FGFRs play a pivotal role acting as cross talkers [91]. When FGFR is muted or epigenetically altered, it can upregulate FGF expression in healthy and tumor cells, which can exert both autocrine and paracrine signaling, generating positive feedback with an impact on tumor progression, cancer cell survival, proliferation, angiogenesis, invasion, metastatic dissemination and response to therapy, due to an amplification of the physiologic growth factor effect of FGFs [91,92,93]. Further investigation is required to elucidate the consequences of genetic alterations in FGFR4 and epigenetic alterations in the remaining liver following PH of tumors as well as in livers grafts implanted in the recipient after removing liver with HCC. If this is the case, strategies aimed exclusively to regulate FGF15/19 levels might not be enough to protect in such surgical conditions.

The role of FGFRs on metastatic dissemination has been explained by their interaction with cell adhesion molecules and activated downstream signaling [54]. Actually, overexpression of FGFR4 in hepatocarcinoma cells has been related to lower cell adhesion and more anchorage-independent growth (in vitro) and lymph/blood-endothelial barrier disintegration, all three key steps in the metastatic process [94]. Moreover, FGFR4 has been identified as a key intermediary in PI3K/AKT and phospholipase C gamma (PLCγ)-mediated metastasis [95,96]. The potential role of the mitogen-activated protein kinase kinase/extracellular signal-regulated kinase (MEK/ERK) pathway mediating the effects of FGFR4 on metastasis should not be averted. Indeed, MEK/ERK pathway has been reported to be activated by FGFR4 [97]. In addition, several works indicated the involvement of the sustained signal transduction of the MEK/ERK pathway in the effects of FGFR1 promoting the development of metastasis [54]. To support this hypothesis, the following data should be considered. Ras/Raf/MEK/ERK pathway has been detected continuously active in HCC following the interaction with epidermal growth factor receptor (EGFR), tyrosine-protein kinase Met (c-Met), and FGFRs [98]. The hepatocyte growth factor/c-Met (HGF/c-Met) pathway is a key mediator in the mesenchymal-epithelial transition, a process required for the invasion of blood or lymph vessels by cancer cells as a preliminary step to migrate cancer cells to distant organs [54]. Similar to EGFR, FGFR4 can interact with VEGFR and platelet-derived growth factor receptor (PDGFR), which via PI3K/AKT/mTOR (mechanistic target of rapamycin), mediates metastatic and angiogenesis processes (Figure 1) [28].

Finally, FGFR4 can bind to neuronal cell adhesion molecule (NCAM), which results in the activation of the downstream signaling pathway, PLCγ, which is related to cell migration, invasion, and metastasis [99]. Moreover, FGFR4 and NCAM interaction can stimulate β1-integrin-mediated cell-matrix adhesion and induce changes in tissue architecture [100].

Given the data explained above, the mechanisms involved in the metastatic process mediated by FGFR4 seem to be similar to those mentioned in relation to tumorigenesis and its effect on PH and LT. Moreover, remaining undetectable micrometastasis after PH in a liver cancer treatment context may be a major problem to achieve total remission of the disease [62].

Table 1 summarizes the main preclinical experimental models that have been used to discuss both tumorigenesis and liver regeneration sections from this review.

## 3. Role of FGF15/19 in Liver Regeneration

The liver is the only organ in mammals that can fully regenerate after several injuries, including PH of more than 70% of adult liver mass [35,110]. Regeneration is a very complex process that involves multiple organs, several types of signaling networks and mobilizes liver stellate cells, liver sinusoidal endothelial cells, and liver stem cells, apart from hepatocytes, to successfully accomplish hepatic regeneration and repair [21,111]. After PH, a coordinated replication of all liver cell types begins, starting with hepatocytes and ending with the rest of non-parenchymal cells. This cellular regeneration is accompanied and guided by many signals, both humoral and intracellular [112,113]. However, the regenerative process is compromised in PH and LT, especially in steatotic livers, since IR affect negatively liver regeneration [21]. One of these first induced hormonal signals after PH is FGF19. Just after PH, a rapid but transient BA overload is present in the liver. BA reabsorbed by enterocytes induces FXR and FGF19 secretion. Through portal circulation, FGF19 reaches the liver where, after binding to the FGFR4-β-klotho complex on the hepatocyte membrane, initiates the proliferative signaling (Figure 2) [35]. The hepatic proliferative characteristics of FGF19 are in line with the fact that FGFR4 is involved in liver architecture, promotes protein synthesis [114,115], and prevents fibrosis upon liver insults [31].

### 3.1. Involvement of BAs and YAP in the Effects of FGF15/19

Over the last years, and thanks to data from different animal and cellular models, many details of the role of FGF19 on liver regeneration have been elucidated, especially in the surgery of PH and LT. Indeed, data from our group on the protective capacity of FGF15 in LT from BD donors indicated that when regeneration response is activated after the appearance of liver damage, some hepatic progenitor cells can be detected. We were able to prove both that these hepatocyte-like progenitor cells are, in fact, hepatocytes and that FGF15 is associated with the formation of these progenitor cells [100]. In another work, Kong et al. using FGF15 transgenic mice overexpressing FGF15 indicate that FGF15 reduces BA synthesis and may directly promote cell proliferation [103]. However, the exact molecular mechanisms and pathways that orchestrate initiation, promotion, and termination of regeneration after PH to effectively restore liver mass remain incompletely understood, and future studies are still necessary to clarify these underlying mechanisms.

Therefore, what do we know to date of FGF15/19′s involvement in regeneration? It can be summarized in two main roles that interact with each other: the stabilization of BA homeostasis to prevent BA accumulation in the regenerating liver that can damage hepatocytes and its participation in the regenerative initiation mechanisms as it has been reported in PH and LT [105].

Several studies highlight the importance of metabolic signals in liver regeneration after PH [116,117]. BAs regulate metabolic homeostasis, tumorigenesis, and immunity. In the same line, different studies indicate that BA induces hepatic proliferation [89,95,111,118,119,120]. Nevertheless, BAs are very toxic substances, especially for the regenerating liver. In line with that suggested by other authors [35], in our view, BA might trigger signaling pathways aimed at promoting hepatic regeneration, but under exacerbated BA accumulation, hepatic BA levels need to be drastically reduced to avoid its hepatic injurious effects. BA synthesis is reduced at the beginning of post-PH liver regeneration [121] once the proliferative signals have been relayed by hepatocytes in order to repair damaged cells [35]. Evidence of the importance of FXR signaling pathway on BA detoxification is supported by the following experimental results: first, FXR deficiency enhances mortality and delays liver regeneration after PH [122]; second, increment in intrahepatic BAs can cause continuous cell damage and necrosis [101] and, finally, FXR absence and hepatic damage injury may cause hepatocyte apoptosis [123]. Nevertheless, in experimental models of PH, FXR deletion has not shown a complete blockade of liver regeneration [124], a fact that may indicate that there exist other compensatory mechanisms.

In LT from BD donor, the reduction in hepatic FGF15 levels after BD was related to a decrement in *yap* gene expression and with alterations in the Hippo-YAP pathway, and this was reflected in hepatic regenerative failure [20]. The treatment with exogenous FGF15 regulated the Hippo-YAP pathway, thus promoting liver regeneration. These results are in accordance with other studies indicating that YAP mediates cell proliferation and participates in the restoration of the heart after IR; it is a key regulator of organ size and can be identified as one of the main regulators of hepatic cell proliferation [125,126,127]. Thus, FGF15/19 mediated by the Hippo-YAP pathway can control regeneration. However, the inhibition of FGF15 does not induce changes to the Hippo-YAP pathway [100]. Thus, depending on the pathological conditions and surgical setting, the proliferative response seems to be not only dependent on Hippo-YAP.

### 3.2. Signaling Pathways (Different to BAs and YAP) Regulated by FGF15/19

It has been reported that the lack of hepatic FGFR4 activates regeneration compensatory pathways. In addition, under the presence of FGF15 other proteins different to YAP related to cell proliferation like Cyclin D1 or Cyclin E are upregulated [86,100]. In this line, it is important to consider that FGFR4 can mediate proliferation through ERK1/2 and PI3K pathways [88,128]. In addition to ERK1/2 and PI3K, MAPK and Stat3 have also been involved in liver regeneration in the PH model. MAPK mediates BA detoxification and ERK1/2 activation, whereas signal transducer and activator of transcription 3 (Stat3)-by forkhead box protein M1 (FOXM1) activation- is involved in the G0-G1 cell cycle transition [86,100]. Both pathways (MAPK and Stat3) are well-known mitogenic drivers that participate in the priming of liver regeneration. Indeed, in a 2018 study, Kong et al. in a model of PH in mice described that MAPK and Stat3 activation are directly induced by FGF15 (Figure 2). Moreover, FGF15 is responsible for the activation of NF-κB, another mediator participating in the first stages of liver regeneration in mice [86,100,129]. A summary of the main preclinical models used to comment this section can be found in Table 1.

Unraveling the exact mechanism of FGF15/19 and its downstream signaling pathways, it is of great importance to elucidate the exact mechanism governing the initiation and termination of proliferative signaling in the liver. This question will be solved by future investigations focused on FGF15/19, and its downstream signaling pathways are given the role of FGF15/19 promoting controlled liver regeneration or participating in the deregulation of the hepatic cell proliferation as occurs in tumorigenesis. This will clarify in detail the paradoxical role of FGF15/19 (that can repair a damaged liver or promote tumorigenesis). Our hypothesis is that, in reality, this dual role of FGF15/19 is logical if its involvement in regeneration processes is taken into account. A molecule that is a growth factor whose function is to promote regeneration may be counterproductive if it is activated or intervened in tumor cells because its function will remain the same, promote proliferation (in a healthy cell), but in a tumor context, this is the same as saying *promote tumorigenesis* since the proliferation promotion is occurring in a tumor cell. This needs to be studied, keeping in mind that in liver surgeries, PH is applied to remove the tumor, but as mentioned above, undetectable micrometastases may remain in the remaining liver following PH. In these circumstances, to reach to standard liver size, the remaining liver needs to be regenerated, but this might induce negative effects in the areas where micrometastases are present in the remaining liver following PH, which would promote tumorigenesis rather than a regenerative process to reach the standard liver size, with the risk of cancer recurrence that this entails.

In summary, the regulation of FGF15/19 might be a promising approach in liver surgery due to its beneficial effects on damage and regenerative failure. On the other hand, as stated, most of the mechanisms controlled by FGF15/19 and involved in liver regeneration are shared by tumorigenic processes. Therefore, it is crucial the establishment of pharmacological or surgical interventions based on FGF15/19 physiological implications in PH and LT without pro-tumorigenic effects to avoid the added risk of tumorigenesis. As we will explain below, the use of non-tumorigenic variants of FGF15/19 may be an appropriate option.

## 4. Modulation of FGF15/19 Actions

### 4.1. Non-Protumorigenic Variants of FGF15/19

With the aim of avoiding the potential pro-tumorigenic activity of FGF19 and in order to evaluate the beneficial properties of FGFR signaling in different liver diseases, non-pro-tumorigenic variants have been studied. This is the case of aldafermin (NGM282), a drug that shows positive results in the treatment of liver diseases as non-alcoholic steatohepatitis (NASH) [130] or primary sclerosing cholangitis [131]. Other studies support the idea that FGF15 could be used as a potential therapeutic option in other liver diseases [132,133]. Studies reported by our group indicated that a single dose of FGF15 in BD and cardiac death (CD) donors promote less damage and regenerative failure in liver grafts without observing hepatic tumor signals [20,21]. Nevertheless, these studies did not evaluate the long-term effects of FGF15. In our view, in order to avoid the possible long-term oncogenic risks, the treatment with non-pro-tumorigenic variants would be the best therapeutic option in the treatment of steatotic and non-steatotic liver grafts used for LT. In any case, more clinical and preclinical studies need to be performed in order to evaluate if these non-oncogenic variants present the same beneficial properties in LT surgeries.

Due to the importance of FXR pathways in the development of various liver diseases, as has been previously mentioned in the current review, not only variants of FGF15/19 have been developed. Thus, many drugs as inhibitors or antagonists of FXR or of the enzymes that belong to the different pathways activated by it are being used in the clinics and studied in preclinical models and randomized clinical trials [36,83] (Table 2). Regarding the FGF19 variant aldafermin, as it has been said before, it presents positive results in the treatment of NASH, reducing liver fat and producing a trend toward fibrosis improvement in patients submitted to phase 2 clinical trials [130]. As it has been said previously, NGM282 also presents positive results in the treatment of primary sclerosing cholangitis [131]. The molecule known as fibapo is a variant developed by Alvarez-Sola et al., and it consists of a fusion molecule encompassing FGF19 and apolipoprotein A-I. A preclinical study demonstrated that the variant fibapo reduced liver lipid, BA accumulation and inhibited endoplasmic reticulum (ER) stress, improving fatty liver regeneration in mice submitted to hepatic resections without detecting tumorigenic signals [134]. Thus, the variant fibapo improved liver regeneration. Zhou et al. engineered a non-tumorigenic variant called M70 that differs from FGF19 in N-terminus, the region involved in receptor interaction and signaling modulation. M70 maintains the ability to repress CYP7A1 expression but no longer triggers activation of STAT3, a signaling pathway essential for FGF19-mediated hepatocarcinogenesis [135]. However, it should be considered that STAT3 is crucial to promote liver regeneration in PH and LT [136,137]. On the other hand, even though STAT3 is widely known as a promoter of liver regeneration, a recent study points out that STAT3 deficiency also promotes biliary proliferation and avoids HCC. This fact could explain why its inhibition may inhibit HCC while avoids liver injury [138]. Thus, the effects of this FGF19 variant, M70, in the surgery setting remain to be elucidated for the potentially injurious effects on the regenerative process. The use of FGF-like molecules that avoids HCC by inhibiting the activation of pro-proliferative pathways, such as M70, could affect regeneration, and therefore more research and clinical trials are needed. However, the use of drugs such as fibapo, which avoids liver injury by modulating metabolism [134], could be the best approach since it does not directly affect regeneration.

### 4.2. Other Drugs That Affect FGF15/19 Signaling Pathways

In addition to these non-protumorigenic variants of FGF15/19, there are some drugs that affect FGF15/19 signaling pathways, including different kinases. For instance, inhibitors or antagonists of its receptor FXR are now being used in clinic and preclinic studies with the aim of treat oncogenic diseases. Sorafenib is one of the principal therapeutic options inside FGF-related drugs in cases of unresectable HCC [139,140] as well as lenvatinib [141,142], regorafenib [141,143], or cabozantinib [141] in cases of resistance to the firsts drug. Drugs such as brivanib [144], dasatinib [145], everolimus [146,147,148], or LD-1 [83] are other therapies affecting FGF15/19 pathways used to treat HCC or other diseases. These data show that despite the most of these drugs act affecting the same pathways and targets, they can have different efficacies and effects, demonstrating that more research, preclinical studies, and randomized clinical trials are required in order to continue exploring the therapeutic uses of such no specific drugs regulating FGF15/19. Moreover, their effects on PH and LT should be elucidated because the pathways regulated by these drugs play protective roles in damage and regeneration in major liver surgeries [22,149].

BAs exert an important role in many crucial metabolic pathways [150], and they could be explored as potential therapeutical options due to their strong relationship with FGF15/19 function. Despite this important function with the main objective of avoiding liver injury, some studies report the effects of modulating their absorption or even their synthesis. Some studies reported the effects of inhibiting or diminishing BA production. Most of these studies focus their efforts on the inhibition of cholesterol 7-alpha hydroxylase [151,152,153]. On the one hand, it has been described that ketoconazole may reduce levels of BAs drastically. The studies reported a high decrease in BAs synthesis in in vivo and in vitro models modulated by fibrates [153] and ketoconazole [151,152]. Even though this important fact has been reported, the liver regenerative ability or liver injury has not been assessed in these contexts. It could be of clinical and scientific interest the study of BAs synthesis pharmacological modulation in liver damage associated with either PH or LT surgeries. On the other hand, Anna Baghdasaryan et al. study reported the positive effects of avoiding their absorption. They describe an improvement of cholestatic liver and bile duct injury [154]. Nevertheless, due to the beneficial and necessary properties of some BAs, avoiding their production entirely may provoke a negative effect despite the potential benefit regarding liver injury. In fact, there exist many disorders regarding the synthesis of BAs that may cause cholestatic liver disease or even progressive neurological disorders [155]. However, their potential use as therapeutic targets could be further explored in the context of hepatic injury in major liver surgeries in the presence of tumorigenesis since they may be used as a potential target to treat hyperglycemia or even fatty liver disease [156].

## 5. Conclusions and Future Perspectives

The benefits of exogenous FGF15 administration in liver donors with steatotic and non-steatotic grafts, as well as the benefits of FGF15/19 inhibitors in patients with HCC, have been reported. These controversial effects of FGF15/19 depend on the type of pathology and the surgical conditions. This reveals the difficulty of managing a liver disease targeting FGF15/19 or its downstream signaling pathways or receptors: if the same pharmacological strategies are applied indiscriminately to different liver pathologies in patients with different liver phenotypes (steatotic or non-steatotic) or different pathology (surgery of LT and PH or in HCC), the effects may be very different.

Currently, FGF19 variants or drugs that target some elements of the FGF15/19 pathway are being successfully used in the clinical practice of HCC, and others are now being investigated in clinical trials. The potential applications of drugs that specifically regulate FGF15 signaling are numerous in liver surgery, which in turn can lead to increasing the number of organs suitable for LT and may provide a novel therapeutic approach to hepatic resection of tumors. Nevertheless, the use of FGF15 analogs in cases of patients with HCC or other hepatic tumors needs to be explored due to the already known pro-tumorigenic properties of this growth factor. Some of the FGF19 variants do not seem to potentiate proliferative and oncogenic pathways, such as the variants M70, fibapo, or aldafermin. In the field of hepatic surgeries of LT and PH with or without IR, these variants need to be explored yet and the possible long-term effects too. The different types of FGF variants and receptors involved in each pathology or situation should be assessed in detail because its potential use in liver surgeries of LT or PH is poorly explored. The literature in the field is very limited, and the actions of such drugs might be different depending on the surgical conditions (PH vs. LT), type of liver (steatotic versus non-steatotic liver), and donor (BD versus CD donors). In addition, the effects of exogenous FGF15 in clinical trials should be evaluated as well as the long-term effects of non-protumorigenic variants in preclinical models and in randomized control trials, and different factors as donor characteristics (steatotic or non-steatotic liver or BD or CD donors) should be evaluated to clearly define the action of the factor in different pathologies and profiles.

FGF19-FGFR4 axis exerts proliferative and anti-apoptotic activities due to the activation of two major signaling pathways: Ras-Raf-ERK1/2-MAPK and PI3K-AKT and regulation of inflammatory cytokines (TNF-α, IL-1, IL-6, IL-10, CCL2, or M-CSF, among others) in steatotic and non-steatotic livers undergoing LT or PH. Nevertheless, these pathways are also activated in tumor cells, promoting HCC progression. FGF15/19 and FGFR4 action in the liver activating these signaling pathways. This may promote protection against IR damage and regenerative failure. However, at the same time, it might promote tumorigenic processes. This is why FGF15/19 and/or FGFR4 mechanisms of action and their potential therapeutic activity should be further investigated. The dual faces of the FGF15/19-FGFR4 axis make it difficult to discern whether we should inhibit or activate the FGF19/FGFR4 signaling pathway to protect against tumorigenesis and to reduce IR damage and regenerative failure in PH and LT. Genetic alterations in FGFR4 and the effect of epigenetic alteration in the liver surgery of tumors should be investigated in order to elucidate its contribution to liver failure. Thus, in our view, the regulation of the FGF15/19-FGFR4 axis might be a promising approach in hepatic resections and LT due to its beneficial effects on damage and regenerative failure. However, as most of the mechanisms controlled by FGF15/19 and involved in liver regeneration are shared by tumorigenic processes, the use and development of non-tumorigenic variants of FGF15/19 should be explored as a potential therapeutic option for various liver surgeries or diseases. Given the duality of FGF15/19-FGFR4, promoting protection against regenerative failure but promoting the process of tumorigenesis, the establishment of FGF15/19 and/or FGFR4 levels as prognostic factors in the surgery of PH and LT is difficult.

## Figures and Tables

**Figure 1 cells-10-01421-f001:**
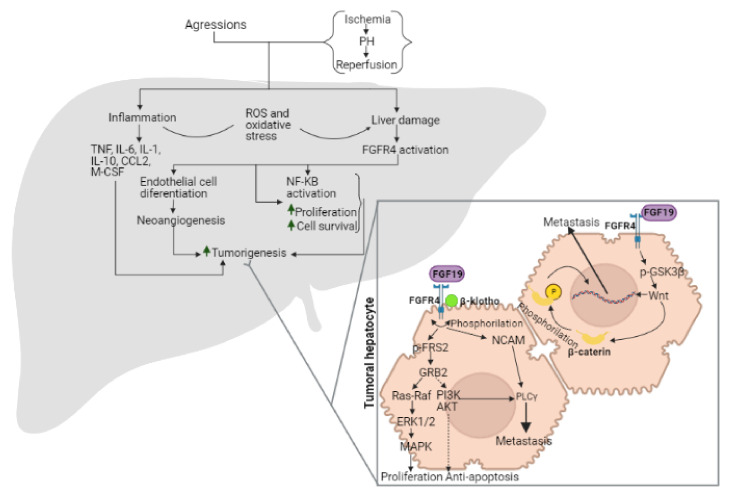
Development of liver tumorigenesis. Tumorigenesis from aggressions and/or tumor resections in the liver, from FGF19-mediated FGFR4 activation and associated inflammatory processes. TNF: tumor necrosis factor; IL-6: interleukin 6; IL-1: interleukin 1; Il-10: interleukin 10; CCL2: chemokine ligand 2; M-CSF: macrophage colony-stimulating factor; ROS: reactive oxygen species; FGFR4: fibroblast growth factor receptor 4; NF-KB: nuclear factor kappa-light-chain-enhancer of activated B cells; P-FRS2: phospho-fibroblast growth factor receptor substrate 2; GRB2: growth factor receptor-bound protein 2; NCAM: natural cell adhesion molecule; PLCγ: phosphoinositide phospholipase C-γ; PI3K: phosphoinositide 3-kinase; AKT: protein kinase B; ERK1/2: extracellular signal-regulated protein kinases 1 and 2; MAPK: mitogen-activated protein kinases; FGF19: fibroblast growth factor 19; p-GSK3β: glycogen synthase kinase 3 beta.

**Figure 2 cells-10-01421-f002:**
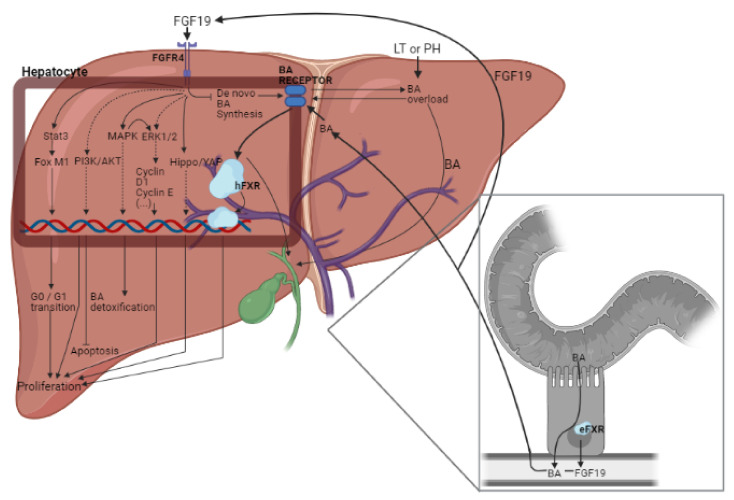
Liver regeneration process mediated by BAs. FGF19 is secreted by enterocytes post-LT or PH in response to transient BAs overdose. When FGF19 reaches the liver via the portal vein, it binds to FGFR4 and mediates proliferation signals. FGF19: fibroblast growth factor 19; FGFR4: fibroblast growth factor 4; Stat3: signal transducer and activator of transcription 3; Fox M1: forkhead box M1; PI3K/AKT: phosphoinositide 3-kinase/protein kinase B; MAPK: mitogen-activated protein kinases; ERK1/2: extracellular signal-regulated protein kinases 1 and 2; Hippo/YAP pathway; hFXR: hepatic farnesoid X receptor; BA: bile acid; LT: liver transplantation; PH: partial hepatectomy; eFXR: enterocyte farnesoid X receptor.

**Table 1 cells-10-01421-t001:** Table summarizing the main preclinical models used to study the processes that drive liver tumorigenesis or regeneration.

Type of Model	Type of Sample	Treatments Applied Based on FGF15/19-FGR4 Axis Modulation	Cancer Induction	Purpose to Evaluate	Reference
Human BD donors	Human biopsy	Any treatment	No	Cell types expressing FGF15 and FGFR4	[20]
Mouse immortalized hepatocytes (AML12), Hepa1-6 hepatoma cells, and C2C12 myoblasts	Cell culture	Administration of lipid nanoparticles carrying FGFR4 siRNA for gen expression blockage	No	FGFR function in liver regeneration	[101]
JHH4, HEP3B, JHH7, HUH7, PLC/PRF/5, and JHH5 HCC cells	Human cell culture	Incubation with anti-FGFR4 monoclonal antibody (LD1)	Yes	FGFRs expression in liver cancer, FGFR4 participation in colony formation, and evaluation of therapeutic potential of LD1 to inhibit FGFR4 function in cancer	[83]
BaF3 pro-B cells IL-3-dependent	Human cell culture	FGFR4 chimeric construct transfection and incubation in the absence of IL-3	No	FGFR4 pro-mitogenic capabilities	[83]
Primary rat hepatocytes	Cell culture	Incubation with human FGF21 and FGF19 and mouse FGF15	No	To determine FGF19 and FGF15 functions	[102]
Normal and FGFR4^−/−^ mouse liver tissue, DEN-initiated hepatomas and derived hepatoma cells	Cell culture from hepatoma samples	Transfection with full-length murine βKL and incubation with FGF19 or FGF1	Yes	βKL role in FGF19 or FGF1-mediated FGFR4 function	[31]
Hepatocytes isolated from 70% PH non-steatotic mice without ischemic period	Cell culture	Incubation whit siRNA of FGFR4 and recombinant human FGF19	No	FGF19 role in LR after PH	[22]
LT of steatotic and non-steatotic BD donor rat liver grafts	SD and ZKob/ob male rats	Administration of FGF15 alone or combined with BA or YAP inhibitor	No	Effects and signaling pathways implication of FGF15	[20]
LT of steatotic and non-steatotic CDD rat liver grafts	ZKob/ob and ZKob/− male rats	FGFR4 inhibitor in donors	No	Role of FGF15 and signaling pathways	[21]
Tissue-specific inducible FGF15 Tg mice undergoing 70% PH without ischemia	C57BL/6J mice	Administration of Dox to *fgf15* transgene inhibition	No	Actions of FGF15 on LR	[103]
Whole-body Fgf15 KO mice undergoing 70% PH without ischemia	C57BL/6J mice	AAV-FGF15 overexpressing FGF15	No	Effects of FGF15 on LR	[103]
Rats subjected to 80% PH with IR	SD male rats	Any treatment	No	Effects of warm IR ** caused by LT on LR	[104]
FGF15 KO mice undergoing 70% PH without ischemia	75% C57BL/6J and 25% 129SvJ mice *	Any treatment	No	FGF15 actions on LR	[105]
FGF15^−/−^ mice subjected to HCC induction by DEN + CCl_4_ administration	C57BL/6129/Sv mice *	Any treatment	Yes	FGF15 role in HCC	[85]
Whole-body FXR KO, intestine-specific and liver-specific FXR null in mice undergoing 70% PH or CCl_4_ liver injury induction	-	Any treatment	No	Restorative mechanisms of FGF15 in LR	[106]
Hepatocyte-specific FGFR1 and FGFR2 KO mice undergoing 70% PH	-	FGFR4 siRNA	No	Impact of FGFR4 loss in LR	[101]
Wild-type mice	FVB strain female mice	ID1 antibody and FGF19 administration	Yes	Effects of FGFR4 inhibition on HCC	[83]
Xenograft mice	nu/nu female mice	5 × 106 mice cancer cells inoculation and LD1 administration	Yes	Effects of FGFR4 inhibition on tumor growth	[83]
FGF19 Tg and simultaneously FGFR4 KO mice	Progeny of a breed FGF19 Tg with FGFR4 KO mice	FGF19 Tg mice	Yes	FGFR4 role in HCC development	[83]
Wild-type mice undergoing 70% PH	C57BL/6 male mice	Any treatment	No	BA flux	[107]
Rats with a biliary fistula with or without chemical compound administration (CCl_4_ and meloxicam) administration	SD male rats	Any treatment	No	Implication of enterohepatic circulation of BA in the outcome of LR	[107]
db/db mice	C57BL6/ JbomTac-KS male mice	Any treatment recombinant human FGF21, human FGF19 or mouse FGF15	No	Actions of both mouse FGF15 and human FGF19	[102]
C57BL/6 Tg mice	C57BL/6 hepatocyte-specific Fgfr4 KO and Frs2α floxed male mice	Conditional Frs2α ablation	No	Mechanisms by which FGFR4 regulates BAs	[108]
FGF15 KO mice undergoing 70% or 85% PH	C57BL/6/129/Sv * mice	AAV-Fgf15 injection and 2% cholestyramine resin dietary administration	No	FGF15 role in BA homeostasis	[109]
Wild-type mice undergoing 70% PH	C57BL/6 male mice	Any treatment	No	FGF15-FGFR4 axis role in LR	[22]

SD: Sprague–Dawley; ZK: Zucker; BD: brain death; BA: bile acid; CDD: cardiac death donor; βKL: β-klotho; IL-3: interleukin 3; Tg: transgenic; PH: partial hepatectomy; LT: liver transplantation; LR: liver regeneration; AAV: adeno-associated virus; KO: knock-out; IR: ischemia reperfusion; FGF15/19: fibroblast growth factor 15 or 19; FGFR4: fibroblast growth factor receptor 4; HCC: hepatocellular carcinoma; YAP: Yes-associated protein; CCl_4_: carbon tetrachloride; FXR: farnesoid X receptor; Frs2α: fibroblast growth factor receptor substrate 2α. * This particular genetic background is used because a C57BL/6J-FGF15 KO is embryonically lethal. ** Warm ischemia: in transplantation procedures, it is a period of time where the irrigation of an organ is blocked but the organ is still in the body and therefore at physiological temperature.

**Table 2 cells-10-01421-t002:** Table summarizing the main FGF15/19-related drugs explaining the action mechanism and the disease where it is used.

Drug	Action Mechanism	Disease
Aldafermin (NGM282) [130,131]	FGF15 variant. Activation of the FGFR1c-KLB receptor	Cholestatic liver disease and NASH (clinical trials)
Fibapo [134]	Interaction with scavenger receptor class B type I (SR-BI)	Fatty liver regeneration (Preclinical model FGF15^−/−^ mice)
M70 [135]	Repression of Cyp7a1 expression but not STAT3 activation	Steatohepatitis and fibrosis
Brivanib [144]	Tyrosine kinase inhibitor and FGFRs inhibitor	HCC
Dasatinib [145]	Tyrosine kinase inhibitor	Some kind of leukemias
Sorafenib [139,140]	Multi-kinase inhibitor	HCC and other cancers
Everolimus [146,147,148]	mTOR inhibitor	Immunosuppressive drug or treatment of some cancers
Lenvatinib [141,142]	Inhibitor of multiple receptor tyrosine kinases	Unresectable HCC and some other cancers
Regorafenib [141,143]	Multi-kinase inhibitor that targets angiogenic, stromal (FGFR), and oncogenic receptor tyrosine kinases	Advanced HCC in patients previously treated with Sorafenib and metastatic colorectal cancer
Cabozantinib [141]	Multi-receptor tyrosine kinase (RTK) inhibitor	HCC resistant to sorafenib
LD-1 [83]	anti-FGFR4 monoclonal antibody	Preclinical model of liver cancer (mice)

NASH: Non-Alcoholic Steatohepatitis; HCC: Hepatocarcinoma.

## Data Availability

Not applicable.

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
