# Peer review of "Role of FGF15 in Hepatic Surgery in the Presence of Tumorigenesis: Dr. Jekyll or Mr. Hyde?"

_cells, 2021, doi:10.3390/cells10061421_

Round 1

Reviewer 1 Report

The authors have built the review around a suggested paradoxical role of FGF15/19 where FGF15/19 displays “controversial effects FGF15/19 in the liver - to repair a damaged liver or to promote tumorigenesis”. However, as alluded to lines 443-452, liver regeneration requires hepatocyte dedifferentiation and proliferation through activation of pathways which might be shared with the tumorigenic process. It is thus unclear what the authors meant to stress out in their manuscript.

Overall, while comprehensive, the review is not well organized and, in many instances, difficult to read. The different ideas need to be better organized and hierarchized and the overall main message needs to be redefined as outlined hereabove.

Also discussed, some specific concepts would benefit from a more straightforward and clear discussion. For example, how FGFR4 regulates metabolism and proliferation in regeneration and tumorigenesis and how these activities are interrelated (for instance BA could regulate liver regeneration through FXR) or can be discriminated needs to be better emphasize.

In section 4, “variants” is used to define drugs derived from FGF15/19 ?

Author Response

Point-by-point response

Reviewer 1 Comments

We thank the Reviewer 1 for your considerations and corrections.

-The authors have built the review around a suggested paradoxical role of FGF15/19 where FGF15/19 displays “controversial effects FGF15/19 in the liver - to repair a damaged liver or to promote tumorigenesis”. However, as alluded to lines 443-452, liver regeneration requires hepatocyte dedifferentiation and proliferation through activation of pathways which might be shared with the tumorigenic process. It is thus unclear what the authors meant to stress out in their manuscript.

Given your comments, this point has been clarified. Please, see pages 14-15.

-Overall, while comprehensive, the review is not well organized and, in many instances, difficult to read. The different ideas need to be better organized and hierarchized and the overall main message needs to be redefined as outlined hereabove.

Your concerns have been taken into in account. The revised version of the review has been organized in different sections. The different ideas have been better organized and hierarchized along the manuscript and the main message has been redefined. Please, see the added subsections between pages 2 and 17.

-Also discussed, some specific concepts would benefit from a more straightforward and clear discussion. For example, how FGFR4 regulates metabolism and proliferation in regeneration and tumorigenesis and how these activities are interrelated (for instance BA could regulate liver regeneration through FXR) or can be discriminated needs to be better emphasize.

This has been considered and included in the revised version of this review. Please, see pages 2-5 and 14-15.

-In section 4, “variants” is used to define drugs derived from FGF15/19?

This has been clarified in the revised version of this manuscript. Please, see page 2.

Reviewer 2 Report

Caballeria-Casals et al. describe the role of FGF15/19 in hepatic surgery in the presence of tumorigenesis which is of great relevance for clinic. They discuss promising strategies of the beneficial use of this pathway accounting its protective effects against liver damage, preventing regenerative failure in LT from BD donors and take into account the pro-tumorigenic activity of FGF15/19 and possible alternatives. Preclinical models and potential pharmacological use are debated, and authors consider non-pro-tumorigenic variants for the treatment of liver diseases in the surgery of hepatic resections and LT.

The review is of great interest adding new views and instigating new ideas that could advance the field with the ultimate purpose of treatment and preventing liver diseases.  

Major comments:

It would be appreciated if authors could elaborate more on:

  1. The role of (FGF)15/19 in the metabolism and how this affects possible outcomes.
  2. Role of BA in tumorigenesis in general and YAP regulation
  3. If the protumorigenic effect of FGF15/19 are avoided using new variants, how does in general affect regeneration? Do the reviewers expect that this new variant would have reduced positive effects in the regeneration?
  4. Do inhibitors of BA synthesis have preventive effects by avoiding injury?

Minor comments:

Line 26/Line 54: Fibroblast growth factor (FGF)15/19

Line 35: Very long sentence, make it two sentences

Line 35: Activation

Line 36: -which, remove –

Line 36: diseases-, remove –

Line 54: FGF19 is secreted by ileal enterocytes

Line 56: Revise reference 10, maybe only 11 is needed

Does the authors have any comments regarding Zhou work:  https://doi.org/10.1016/j.devcel.2018.12.021. Could you comment the controversial findings?

Line 63: Revise reference 13

Line 63: However, of scientific and clinical interest, many studies have reported the negative effects of YAP pathway (activated by FGF15/19) in hepatocarcinoma (HCC) and other tumors (cancers?) [14,15]. Thus FGF15/19 and its downstream signaling pathways have two faces dependently of the pathological conditions.

Line 67: in this manuscript the dual effect of … will be reviewed.

Line 75: disease (s)

Lines 73-75, revise this sentence. These aggressions not always cause all of the spectrum of liver diseases at the same time. Define a little bit better.

Line 79. Shorten the sentence. Full stop after [20]. Since

Line 79. Might promote?

Line 82: regenerative-role properties avoid proliferation of damaged hepatocytes? Elaborate or explain this sentence a little bit better.

Lines 83-86: Rewrite this sentence

Line 87-89: Define what process are enhanced or reduced

Line 88: Reference

Line 90: bile acid synthesis, in intermediary metabolism, and the high

Line 94: change to FAVOR the transformation of …

Line 95: This is consistent with several works pointing out that FGFRs signalling abnormalities observed in liver cancer can contribute to the mechanisms involved in liver cirrhosis [27]. ??

Line 92-101: Rephrase so the paragraph sense is more logic.

Line 100: TNF, IL-1, IL-6, IL-10, CCL2 (abbreviations in the main text) Also NFkB

Line 103: inflammatory microenvironment

Line 108: by activating survival pathways to reduce damage and RESOLVE inflammation [31].

Line 114: some of these IR-mediated CKs might cause hepatocyte death during PH and LT. In addition, IR might not only cause hepatocyte death by a storm of inflammatory mediators, but also other factor might be involved, like oxygen delivery, … Is this also known?

Line 115: such mediators might be released to the circulation of the recipient contributing to tumour progression.

Line 120: In addition, during PH, it should ....undetected and unresected micrometastasis {34}

Line 121; thus, these inflammatory mediators

Line 122-127; promote/prevent hepatic damage depending on the nature (quality) and quantity of the inflammatory mediators?

Can the author elaborate more regarding the quality and quantity of these inflammatory mediators and their outcome?

Line 134: abbreviation, phosphorylated GSK3β

Line 136: That lose/ This loss

Line 143-144: However, THIS EFFECT was abrogated WHEN the FGF19-FGFR4 interaction WAS disrupted using an antibody against anti-FGF19 [37].

Line 145-147: In addition, studies based on mouse models indicate that individual or combined overexpression of FGF19 and FGFR4 are not the only risk factors for HCC and other cancer types, since FGFR4 activation mediated by abnormal FGF19 signalling is just an important issue ???[40].  Clarify this sentence. It is difficult to understand

Line 152: upregulation of FGFR4 expression

Line 177: The crucial role of FGFR4 downstream signalling pathways in reducing IR injury and promoting cell damage repair has been previously demonstrated in steatotic and non-steatotic livers undergoing either PH or LT

Line 184: HCC abbreviated. (all manuscript)

Line 190: anti-proliferative effect

Line 205: In vitro and in vivo in italics

Line 207: HCC-derived cells

Line 226: much more appropriated models (better models)

Line 227: are required

Line 228: that change to the

Line 231: DEN-mediated

Line 270: Obviously,

Line 275: , and a

Line 284: Original reference rather than review. Same for all.

Line 292: alterations,

Line 302: will be also is required

Line 302: Further investigation will be also required to elucidate the potential CONSEQUENCES of genetic alterations in FGFR4 and the contribution of epigenetic alterationS in the remaining liver following PH of tumours as well as in livers grafts implanted in the recipient after removing liver with HCC.

Line 310: lymph

Line 310: in vitro

Lines 312: Abbreviations?

Line 314: discarded Obviated or averted

Lines 320: transition-, remove – and add a comma (,)

In general, include the description of the abbreviations also in the main text, not only in the figure.

Line 332: (see below)

Figure 1: Improve resolution or size of text.

Table1:

  • Therapies applied (separated). I would include in parenthesis (FGF19). I would change therapies to treatments. Some of the treatments used are not therapies.
  • [11] clarify what do you mean with both molecules FGF19 and FGR4
  • Administration of lipid nanoparticles carrying FGFR4 siRNA FOR gen expression BLOCKAGE
  • JHH4 liver cancer cells; are these human cells, mouse cells??
  • Therapeutic (there are 2 spaces) potential
  • Clarify the origin of the cells and if possible, everywhere in the table, include their abbreviations in the legend.
  • Combine information of reference 54 in one line. At least these ones: HEP3B and JHH7 liver cancer cells, HUH7, PLC/PRF/5 and JHH5 HCC cells, JHH4 liver cancer cells
  • Ref 21, are these isolated hepatocytes? How do they culture mouse liver tissue, organoids?
  • BD donor or BDD, maybe better BD donor as it is written like this in all the review
  • Effects abd signalling pathwus of FGF15 [11]. Revise sentence
  • Rats submitted to 80% PH with IR. SUBJECTED instead of submitted [75]
  • Warm IR? What is warm IR?
  • [56] subjected better than submitted
  • There is an extra space between [56] and [77]
  • [77] Reparation restorative on in
  • Hepatocyte-specific Fgfr1 and FGFR2 in KO mice undergoing 70% PH. Both in lowercase? in ??
  • ID1 antibody and FGF19administration. Correct space
  • Overexpressed [54] not needed
  • Progeny of a breEd between FGF19 Tg and FGFR4 KO mice
  • [78] compound. Double check spelling or words and grammar in the table
  • Implication of enterohepatic circulation of BA in the outcome of LR [78]
  • Actions of both mouse FGF15 and human FGF19 [72]
  • FVB?
  • FGFR4 Tg mice and then C57BL/6 hepatocyte-specific Fgfr4 null and Frs2α floxed male mice??? I don’t understand this FGFR4 Tg but also Fgf4 null???
  • Role in [73]
  • Make sure all of the abbreviations are included.
  • Lines 361: One of these first INDUCED hormonal signals after PH is FGF19.
  • Lines 362: overload is present in the liver. BA reabsorbed by enterocytes induce FXR and FGF19 secretion.
  • activation of farnesoid X receptor (FXR), this abbreviation is only necessary at the beginning. Only once.
  • Through portal circulation, FGF19 reaches the liver where,

Line 365: on the hepatocyte mb.

This data

Line 366. These data on the hepatic proliferative characteristics of FGF19 are in line with THE FACT that FGR4 is involved…

brain dead donors, BD donors

Line 376: more consistency with capitals and lowercases through all the manuscript

Tg mice or transgenic mice. If the authors are using abbreviations, this should be consistent through all the manuscript

Line 393: in line with this. (overused sentence)

Line 399: BA, bile acids

Line 402: other compensatory mechanisms of regeneration.

Line 408: BD donor, brain dead donors,

Line 418: It is important to consider

Line 424: mediated induced, or the induction is mediated by

Line 426. A summary of the main preclinical models used to comment this section can be found in Table 1. ()

Figure 2. Improve resolution and/or size of the text inside the figure

BA receptor in singular

Legend: When FGF19 reaches the liver via the portal vein

Lines 437-442: It is known that different pathways initiate and maintain the proliferative signalling in the liver, but the exact mechanisms that promote the initiation and termination need to be exploredThis question will be solved ….

Unraveling the exact mechanism of FGF15/19 and its downstream signaling pathways it is of great importance to elucidate the exact mechanism governing initiation and termination of proliferative signalling in the liver.

Line 443: Finally, given all the data mentioned above and those reported in the section on the 443 role of FGF15/19 in liver tumorigenesis; it should be considered the following observa-444 tions: t In summary, the regulation of FGF15/19…

Line 448: Therefore, the establishment of pharmacological or surgical interventions is crucial based on FGF15/19 physiological implications in PH and LT without pro-tumorigenic effects to avoid the added risks of tumorigenesis.

Line 460: short-term administration of FGF15 in a single dose. Better a single dose of FGF15 in BD and CD…

Line 460: review-, remove –

Line 472: Regarding the FGF19 variant, aldafermin as it has been said before presenting positive results in the treatment of NASH since in a phase 2 clinical trial in patients with NASH, it reduced liver fat and produced a trend toward fibrosis improvement [102]. Rephrase for better understanding

Line 475: NGM282 has it has 475 been said before?

Line 479: lipid, and BA accumulation

Line 500: Against, their effects on PH and LT should be elucidated since such the kinases 500 regulated by such drugs play protective roles in damage and regeneration in PH and LT 501 [74,120]. Rephrase

Line 523: such as the variants…

Line 525-529 too long. Make it two sentences. Full stop after CD donors).

Line 530: FGF15 in separated

Line 539: It is one of the first controversial data in relation to FGF15/19 and FGFR4 actions in the liver because the activate of these signalling pathways and may promote protection against IR damage and regenerative failure. However, at the same time, it might promote tumorigenic processes

Line 547: contribution to what? I would write effect

Line 554: Given the duality

Author Response

Reviewer 2 Comments

We thank the Reviewer 2 for your considerations and corrections.

Major comments:

-The role of (FGF)15/19 in the metabolism and how this affects possible outcomes.

This point has been assessed in page 2.

-Role of BA in tumorigenesis in general and YAP regulation

As you suggest, this point has been assessed in page 3-4.

-If the protumorigenic effect of FGF15/19 are avoided using new variants, how does in general affect regeneration? Do the reviewers expect that this new variant would have reduced positive effects in the regeneration?

More discussion about this fact has been included in Section 4, page 15-16.

-Do inhibitors of BA synthesis have preventive effects by avoiding injury?

Given your comments, this has been added in the last paragraph of Section 4, page 17. 

Minor comments:

-Line 26/Line 54: Fibroblast growth factor (FGF)15/19

It has been corrected. Please, see page 1.

-Line 35: Very long sentence, make it two sentences

The sentence has been divided in two different sentences. Please, see page 1.

-Line 35: Activation

It has been corrected. Please, see page 1.

-Line 36: -which, remove –

It has been corrected. Please, see page 1.

-Line 36: diseases-, remove –

It has been corrected. Please, see page 1.

-Line 54: FGF19 is secreted by ileal enterocytes

It has been corrected. Please, see page 2.

-Line 56: Revise reference 10, maybe only 11 is needed

Reference has been revised and corrected. Please, see page 2.

-Does the authors have any comments regarding Zhou work:  https://doi.org/10.1016/j.devcel.2018.12.021. Could you comment the controversial findings?

Considering your comments, Discussion about Zhou research has been added in the revised version of this manuscript. Please, page 3-4.

-Line 63: Revise reference 13

Reference has been revised. Please, see page 2.

-Line 63: However, of scientific and clinical interest, many studies have reported the negative effects of YAP pathway (activated by FGF15/19) in hepatocarcinoma (HCC) and other tumors (cancers?) [14,15]. Thus FGF15/19 and its downstream signaling pathways have two faces dependently of the pathological conditions.

It has been clarified. Please, see page 2.

-Line 67: in this manuscript the dual effect of … will be reviewed.

It has been corrected. Please, see page 2.

-Line 75: disease (s)

It has been corrected. Please, see page 2.

-Lines 73-75, revise this sentence. These aggressions not always cause all of the spectrum of liver diseases at the same time. Define a little bit better.

This sentence has been revised and corrected. Please, see page 2.

-Line 79. Shorten the sentence. Full stop after [20]. Since

The sentence has been shortened. Please, see page 2.

-Line 79. Might promote?

It has been corrected. Please, see page 2.

-Line 82: regenerative-role properties avoid proliferation of damaged hepatocytes? Elaborate or explain this sentence a little bit better.

It has been explained. Please, see page 2.

-Lines 83-86: Rewrite this sentence

This sentence has been rewrite. Please, see page 2

-Line 87-89: Define what process are enhanced or reduced

The suggestion has been added. Please, see page 3.

-Line 88: Reference

This reference has been added page 3.

-Line 90: bile acid synthesis, in intermediary metabolism, and the high

It has been corrected page 3.

-Line 94: change to FAVOR the transformation of …

It has been corrected. Please, see page 4.

-Line 95: This is consistent with several works pointing out that FGFRs signalling abnormalities observed in liver cancer can contribute to the mechanisms involved in liver cirrhosis [27]. ??

It has been considered and included in the revised version of this manuscript. Please, see page 4.

-Line 92-101: Rephrase so the paragraph sense is more logic.

It has been corrected. Please, see page 4.

-Line 100: TNF, IL-1, IL-6, IL-10, CCL2 (abbreviations in the main text) Also NFkB

It has been corrected (page 4).

-Line 103: inflammatory microenvironment

It has been corrected (page 4).

-Line 108: by activating survival pathways to reduce damage and RESOLVE inflammation [31].

It has been corrected (page 4).

-Line 114: some of these IR-mediated CKs might cause hepatocyte death during PH and LT. In addition, IR might not only cause hepatocyte death by a storm of inflammatory mediators, but also other factor might be involved, like oxygen delivery, … Is this also known?

It has been clarified in the revised version. References have been included. Please, see page 4.

-Line 115: such mediators might be released to the circulation of the recipient contributing to tumour progression.

It has been considered. Please, see page 4.

-Line 120: In addition, during PH, it should ....undetected and unresected micrometastasis {34}

It has been corrected (page 4).

-Line 121; thus, these inflammatory mediators

It has been corrected in page 4.

-Line 122-127; promote/prevent hepatic damage depending on the nature (quality) and quantity of the inflammatory mediators?

Can the author elaborate more regarding the quality and quantity of these inflammatory mediators and their outcome?

This suggestion has been considered and added in the revised version of this manuscript Please, see page 4-5.

-Line 134: abbreviation, phosphorylated GSK3β

It has been corrected (page 5).

-Line 136: That lose/ This loss

It has been corrected (page 5).

-Line 143-144: However, THIS EFFECT was abrogated WHEN the FGF19-FGFR4 interaction WAS disrupted using an antibody against anti-FGF19 [37].

It has been corrected (page 5).

-Line 145-147: In addition, studies based on mouse models indicate that individual or combined overexpression of FGF19 and FGFR4 are not the only risk factors for HCC and other cancer types, since FGFR4 activation mediated by abnormal FGF19 signalling is just an important issue ???[40].  Clarify this sentence. It is difficult to understand

It has been rephrased. Please, see page 5.

-Line 152: upregulation of FGFR4 expression

It has been corrected (page 5).

-Line 177: The crucial role of FGFR4 downstream signalling pathways in reducing IR injury and promoting cell damage repair has been previously demonstrated in steatotic and non-steatotic livers undergoing either PH or LT

It has been corrected (page 6).

-Line 184: HCC abbreviated. (all manuscript)

It has been corrected. See above the manuscript.

-Line 190: anti-proliferative effect

It has been corrected (page 6).

-Line 205: In vitro and in vivo in italics

It has been corrected (page 6).

-Line 207: HCC-derived cells

It has been corrected (page 6).

-Line 226: much more appropriated models (better models)

It has been corrected (page 6).

-Line 227: are required

It has been corrected (page 6).

-Line 228: that change to the

It has been corrected (page 7).

-Line 231: DEN-mediated

It has been corrected (page 7).

-Line 270: Obviously,

It has been corrected (page 7).

-Line 275: , and a

It has been corrected page 8.

-Line 284: Original reference rather than review. Same for all.

It has been corrected (page 8).

-Line 292: alterations,

It has been corrected (page 8).

-Line 302: will be also is required

It has been corrected (page 8).

-Line 302: Further investigation will be also required to elucidate the potential CONSEQUENCES of genetic alterations in FGFR4 and the contribution of epigenetic alterationS in the remaining liver following PH of tumours as well as in livers grafts implanted in the recipient after removing liver with HCC.

It has been corrected (page 8).

-Line 310: lymph

It has been corrected (page 8).

-Line 310: in vitro

It has been corrected (page 8).

-Lines 312: Abbreviations?

It has been corrected (page 8).

-Line 314: discarded Obviated or averted

It has been corrected (page 8).

-Lines 320: transition-, remove – and add a comma (,)

It has been corrected (page 9).

-In general, include the description of the abbreviations also in the main text, not only in the figure.

It has been included as suggested.

Line 332: (see below)

-Figure 1: Improve resolution or size of text.

Resolution of figure 1 has been improved.

-Table1:

Therapies applied (separated). I would include in parenthesis (FGF19). I would change therapies to treatments. Some of the treatments used are not therapies.

It has been corrected (Table 1).

-[11] clarify what do you mean with both molecules FGF19 and FGR4

It has been corrected (Table 1).

-Administration of lipid nanoparticles carrying FGFR4 siRNA FOR gen expression BLOCKAGE

It has been corrected (Table 1).

-JHH4 liver cancer cells; are these human cells, mouse cells??

It has been corrected (Table 1).

-Therapeutic (there are 2 spaces) potential

It has been corrected (Table 1).

-Clarify the origin of the cells and if possible, everywhere in the table, include their abbreviations in the legend.

It has been clarified and the abbreviation has been included as suggested in Table 1.

-Combine information of reference 54 in one line. At least these ones: HEP3B and JHH7 liver cancer cells, HUH7, PLC/PRF/5 and JHH5 HCC cells, JHH4 liver cancer cells

It has been corrected (Table 1).

-Ref 21, are these isolated hepatocytes? How do they culture mouse liver tissue, organoids?

It has been corrected (Table 1).

-BD donor or BDD, maybe better BD donor as it is written like this in all the review

It has been corrected in all review.

-Effects abd signalling pathwus of FGF15 [11]. Revise sentence

It has been corrected (Table 1).

-Rats submitted to 80% PH with IR. SUBJECTED instead of submitted [75]

It has been corrected (Table 1).

-Warm IR? What is warm IR?

It has been explained in page 12 (Table 1).

-[56] subjected better than submitted

It has been corrected (Table 1).

-There is an extra space between [56] and [77]

It has been corrected (Table 1).

-[77] Reparation restorative on in

It has been corrected (Table 1).

-Hepatocyte-specific Fgfr1 and FGFR2 in KO mice undergoing 70% PH. Both in lowercase? in ??

It has been corrected (Table 1).

-ID1 antibody and FGF19administration. Correct space

It has been corrected (Table 1).

-Overexpressed [54] not needed

It has been corrected (Table 1).

-Progeny of a breEd between FGF19 Tg and FGFR4 KO mice

It has been corrected (Table 1).

-[78] compound. Double check spelling or words and grammar in the table

It has been corrected (Table 1).

-Implication of enterohepatic circulation of BA in the outcome of LR [78]

It has been corrected (Table 1).

-Actions of both mouse FGF15 and human FGF19 [72]

It has been corrected (Table 1).

-FVB?

It has been explained in Table 1.

-FGFR4 Tg mice and then C57BL/6 hepatocyte-specific Fgfr4 null and Frs2α floxed male mice??? I don’t understand this FGFR4 Tg but also Fgf4 null???

It has been explained in Table 1.

-Role in [73]

It has been explained in Table 1.

-Make sure all of the abbreviations are included.

It has been revised in all the review.

-Lines 361: One of these first INDUCED hormonal signals after PH is FGF19.

It has been corrected (page 13).

Lines 362: overload is present in the liver. BA reabsorbed by enterocytes induce FXR and FGF19 secretion.

It has been corrected (page 13).

-activation of farnesoid X receptor (FXR), this abbreviation is only necessary at the beginning. Only once.

It has been corrected (page 13).

-Through portal circulation, FGF19 reaches the liver where,

It has been corrected (page 13).

-Line 365: on the hepatocyte mb.

It has been corrected (page 13).

-Line 366. These data on the hepatic proliferative characteristics of FGF19 are in line with THE FACT that FGR4 is involved…

It has been corrected (page 13).

-brain dead donors, BD donors

It has been corrected (page 13).

-Line 376: more consistency with capitals and lowercases through all the manuscript

It has been corrected through all the manuscript

-Tg mice or transgenic mice. If the authors are using abbreviations, this should be consistent through all the manuscript

It has been corrected through all the manuscript

-Line 393: in line with this. (overused sentence)

It has been corrected (page 13).

-Line 399: BA, bile acids

It has been corrected (page 13).

-Line 402: other compensatory mechanisms of regeneration.

It has been corrected (page 13).

-Line 408: BD donor, brain dead donors,

It has been corrected (page 13).

-Line 418: It is important to consider

It has been corrected (page 14).

-Line 424: mediated induced, or the induction is mediated by

It has been corrected (page 14).

-Line 426. A summary of the main preclinical models used to comment this section can be found in Table 1. ()

It has been corrected (page 14).

-Figure 2. Improve resolution and/or size of the text inside the figure

Resolution of figure 2 has been improved (page 14).

-BA receptor in singular

It has been corrected (page 14).

-Legend: When FGF19 reaches the liver via the portal vein

It has been corrected (page 14).

-Lines 437-442: It is known that different pathways initiate and maintain the proliferative signalling in the liver, but the exact mechanisms that promote the initiation and termination need to be exploredThis question will be solved ….

It has been corrected in page 15.

-Unraveling the exact mechanism of FGF15/19 and its downstream signaling pathways it is of great importance to elucidate the exact mechanism governing initiation and termination of proliferative signalling in the liver.

It has been corrected (page 15).

-Line 443: Finally, given all the data mentioned above and those reported in the section on the 443 role of FGF15/19 in liver tumorigenesis; it should be considered the following observa-444 tions: t In summary, the regulation of FGF15/19…

It has been corrected (page 15).

-Line 448: Therefore, the establishment of pharmacological or surgical interventions is crucial based on FGF15/19 physiological implications in PH and LT without pro-tumorigenic effects to avoid the added risks of tumorigenesis.

It has been corrected (page 15).

-Line 460: short-term administration of FGF15 in a single dose. Better a single dose of FGF15 in BD and CD…

It has been corrected (page 15).

-Line 460: review-, remove –

It has been corrected (page 15).

-Line 472: Regarding the FGF19 variant, aldafermin as it has been said before presenting positive results in the treatment of NASH since in a phase 2 clinical trial in patients with NASH, it reduced liver fat and produced a trend toward fibrosis improvement [102]. Rephrase for better understanding

It has been rephrased. Please, see page 15.

-Line 475: NGM282 has it has 475 been said before?

Yes. It has been previously mentioned in the 4th line of section 4 of page 15.

-Line 479: lipid, and BA accumulation

It has been corrected (page 16).

-Line 500: Against, their effects on PH and LT should be elucidated since such the kinases 500 regulated by such drugs play protective roles in damage and regeneration in PH and LT 501 [74,120]. Rephrase

It has been rephrased (page 16).

-Line 523: such as the variants…

It has been corrected (page 18).

-Line 525-529 too long. Make it two sentences. Full stop after CD donors).

The sentence has been divided in two different sentences (page 18).

-Line 530: FGF15 in separated

It has been corrected (page 18).

-Line 539: It is one of the first controversial data in relation to FGF15/19 and FGFR4 actions in the liver because the activate of these signalling pathways and may promote protection against IR damage and regenerative failure. However, at the same time, it might promote tumorigenic processes

It has been corrected (page 18).

-Line 547: contribution to what? I would write effect

It has been corrected (page 18).

-Line 554: Given the duality

It has been corrected (page 18).

Round 2

Reviewer 1 Report

The authors have improved the organization and overall clarity of the main points disussed in the review